# *MCEE* Mutations in an Adult Patient with Parkinson’s Disease, Dementia, Stroke and Elevated Levels of Methylmalonic Acid

**DOI:** 10.3390/ijms20112631

**Published:** 2019-05-29

**Authors:** Mattias Andréasson, Rolf H. Zetterström, Ulrika von Döbeln, Anna Wedell, Per Svenningsson

**Affiliations:** 1Department of Neurology, Karolinska University Hospital, 141 86 Stockholm, Sweden; per.svenningsson@ki.se; 2Department of Clinical Neuroscience, Karolinska Institutet, 171 77 Stockholm, Sweden; 3Center for Neurology, Academic Specialist Center, 113 65 Stockholm, Sweden; 4Center for Inherited Metabolic Diseases, Karolinska University Hospital, 171 76 Stockholm, Sweden; rolf.zetterstrom@ki.se (R.H.Z.); ulrika.dobeln@ki.se (U.v.D.); anna.wedell@ki.se (A.W.); 5Department of Molecular Medicine and Surgery, Karolinska Institutet, 171 76 Stockholm, Sweden

**Keywords:** methylmalonic acidemia, methylmalonic acid, methyl-CoA epimerase, *MCEE* gene, cobalamin

## Abstract

Methylmalonic aciduria (MMA-uria) is seen in several inborn errors of metabolism (IEM) affecting intracellular cobalamin pathways. Methylmalonyl-CoA epimerase (MCE) is an enzyme involved in the mitochondrial cobalamin-dependent pathway generating succinyl-CoA. Homozygous mutations in the corresponding *MCEE* gene have been shown in children to cause MCE deficiency with isolated MMA-uria and a variable clinical phenotype. We describe a 78-year-old man with Parkinson’s disease, dementia and stroke in whom elevated serum levels of methylmalonic acid had been evident for many years. Metabolic work-up revealed intermittent MMA-uria and increased plasma levels of propionyl-carnitine not responsive to treatment with high-dose hydroxycobalamin. Whole genome sequencing was performed, with data analysis targeted towards genes known to cause IEM. Compound heterozygous mutations were identified in the *MCEE* gene, c.139C>T (p.Arg47X) and c.419delA (p.Lys140fs), of which the latter is novel. To our knowledge, this is the first report of an adult patient with *MCEE* mutations and MMA-uria, thus adding novel data to the possible phenotypical spectrum of MCE deficiency. Although clinical implications are uncertain, it can be speculated whether intermittent hyperammonemia during episodes of metabolic stress could have precipitated the patient’s ongoing neurodegeneration attributed to Parkinson’s disease.

## 1. Introduction

Following cellular uptake into the lysosome, cobalamin is converted to adenosylcobalamin (AdoCbl) and methylcobalamin (MeCbl) through two separate pathways. MeCbl functions as a co-factor in the cytosolic methionine cycle and AdoCbl serves as a co-factor for the mitochondrial enzyme methylmalonyl-CoA mutase (MUT) [1,2]. Methylmalonic aciduria (MMA-uria) is present in several inborn errors of metabolism (IEM) affecting different steps of these intracellular cobalamin pathways. In particular, disruption of AdoCbl, and by extension MUT, is known to cause isolated MMA-uria [2]. Stroke affecting the basal ganglia, movement disorders and cognitive decline have been described in the context of isolated MMA-uria [3].

Methylmalonyl-CoA epimerase (MCE) is an enzyme participating, together with MUT, in the mitochondrial pathway metabolizing propionyl-CoA to succinyl-CoA, which in turn enters the citric acid cycle. MCE acts upstream from MUT, converting D-methylmalonyl-CoA to L-methylmalonyl-CoA (Figure 1). Thus, MCE deficiency is also thought to cause isolated MMA-uria [1,4]. 

MCE deficiency, with underlying *MCEE* mutations, has been described in at least 18 children, with phenotypes ranging from acute metabolic acidosis to asymptomatic presentations [4,6,7,8,9,10]. With this case report, we describe an adult patient with MMA-uria and compound heterozygous mutations in the *MCEE* gene, of which one is novel, and discuss possible implications with regard to the patient’s concurrent Parkinson’s disease (PD).

## 2. Case Report

The patient and his two sons have signed informed consents. Patient-related procedures have been approved by the regional ethical board (reference number: 2016/19-31/2, date of approval: 21 March 2016).

The proband is a 78-year-old man of German descent. Family history consists of a mother with dementia at an unknown age. A diagnosis of PD was made in 2004 with satisfactory motor response to dopaminergic treatment. Moreover, the patient has a known medical history of chronic renal failure, cardiomyopathy, hypertension, asthma, prostatic cancer, infratentorial stroke in 2009 and a cortical stroke in 2016. Brain magnetic resonance imaging (MRI) in 2014 demonstrated older basal ganglia infarctions.

In 2007, milder short-term memory difficulties were reported and the patient exhibited a Mini Mental State Examination (MMSE) score of 28/30. Treatment was initiated with memantine but transient episodes of confusion emerged and by the end of 2011, donepezil was added. Cognitive deterioration continued with a Montreal Cognitive Assessment (MoCA) score of 17/30 in 2013. Behavioral symptoms ensued, including hallucinations, anxiety and delusions, causing the patient to be moved to a nursing home. At examination in 2016, the patient exhibited asymmetric parkinsonism with intermittent rest tremor, impaired postural control, a MoCA score of 5/30 and he needed a walker. 

Due to markedly elevated levels of methylmalonic acid (MMA) in serum, a metabolic investigation was undertaken in 2015. First, a treatment trial with hydroxycobalamin categorized the patient’s biochemical profile as a non-responder to cobalamin substitution. Mild MMA-uria, together with increased plasma levels of propionyl-carnitine, were demonstrated in 2016 (Table 1).

Clinical whole genome sequencing ensued, and the subsequent data were analyzed with the Mutation Identification Pipeline (MIP, v3.0.7). The data analysis was in silico, restricted to 681 genes known to cause IEM, as previously described [13]. Two heterozygous mutations were identified in the *MCEE* gene and were validated by Sanger sequencing. The first mutation, c.139C>T (p.Arg47X), is identical to the first described homozygous mutation in MCE deficiency [6]. The second mutation, c.419delA (p.Lys140fs), causes a frameshift introducing a premature stop codon that has not been described before in the context of MCE deficiency (Figure 2).

No significant clinical or biochemical deviations were demonstrated in the proband’s two sons. Sanger sequencing of the *MCEE* gene, in genomic DNA isolated from blood, revealed the sons as heterozygous carriers of one each of the proband’s mutations.

## 3. Discussion

We describe an adult patient with highly elevated plasma MMA and mild MMA-uria likely caused by an underlying MCE deficiency due to compound heterozygous mutations in the *MCEE* gene. Cellular complementation studies and [^14^C] propionate incorporation assays were not performed in this patient. However, one of the mutations, c.139C>T (p.Arg47X), is a nonsense mutation that has been shown to cause MCE deficiency in homozygous state, and the other is a frameshift mutation causing a premature stop codon. Both mutations are thus predicted to result in the synthesis of truncated versions of the MCE protein, supporting an underlying MCE deficiency explaining the patient’s biochemical phenotype [4,6].

The patient demonstrated a modest hyperhomocysteinemia that gradually increased with time and paralleled the patient’s deteriorating renal function. The elevated homocysteine levels persisted despite high-dose substitution with hydroxycobalamin, thereby precluding an underlying cobalamin deficiency (Table 1; footnote ^e^). Thus, the hyperhomocysteinemia was mainly considered to reflect a gradually decreasing glomerular filtration rate.

To our knowledge, this is the first clinico-genetic characterization of an adult with isolated MMA-uria with underlying *MCEE* mutations. We cannot conclude for certain whether the patient’s presumptive MCE deficiency is clinically symptomatic. In theory, increased levels of propionyl-CoA inhibit the enzyme N-acetylglutamate synthase, which is essential for maintaining the urea cycle. In turn, subsequent impairment of the urea cycle could contribute to hyperammonemia which is a known toxic state for the central nervous system [14]. Thus, it can be speculated whether intermittent hyperammonemia during episodes of metabolic stress, and increased loading of MCE-dependent metabolic pathways, might have contributed to or precipitated the patient’s concurrent neurodegeneration attributed to PD. Unfortunately, no testing for hyperammonemia was performed in the described patient. Furthermore, repeated blood gas analysis was not carried out, except in 2014, in which normal pCO2 and lactate levels were seen in venous blood. However, it is notable that the patient already developed cognitive impairment within 3 years after diagnosis of PD, in spite of normal cerebrospinal fluid markers (tau, phosphorylated tau and β-amyloid) in 2012. 

Previous reports of MCE deficiency have included cases with clinically mild phenotypes, which in turn has questioned the clinical relevance of MCE deficiency [10]. Furthermore, the presence of an alternative compensatory metabolic route for D-methylmalonyl-CoA, bypassing the MCE step, has been proposed and thus possibly partly explains the existence of mild phenotypes (Figure 1) [4]. 

## 4. Conclusions

We report a case of MCE deficiency in an adult patient with concurrent PD, stroke and dementia exhibiting highly elevated plasma MMA and mild intermittent MMA-uria. One of the identified underlying compound heterozygous mutations in the *MCEE* gene is novel. Further characterization of phenotypes in adult patients with MCE deficiency are needed in order to better understand possible clinical implications with regard to concurrent neurodegenerative diseases such as PD.

## Figures and Tables

**Figure 1 ijms-20-02631-f001:**
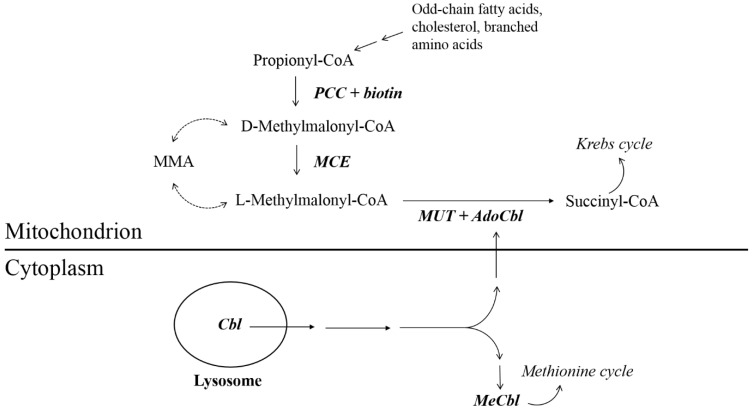
Intracellular cobalamin metabolism. Methylmalonyl-CoA epimerase (MCE) deficiency is thought to generate a block in the metabolic pathway starting with propionyl-CoA. In dotted lines, a possible route bypassing the MCE step is shown [4,5]. Abbreviations: Cbl—cobalamin; MeCbl—methylcobalamin; AdoCbl—adenosylcobalamin; MUT—methylmalonyl-CoA mutase; MMA—methylmalonic acid; MCE—methylmalonyl-CoA epimerase; PCC—propionyl-CoA carboxylase.

**Figure 2 ijms-20-02631-f002:**
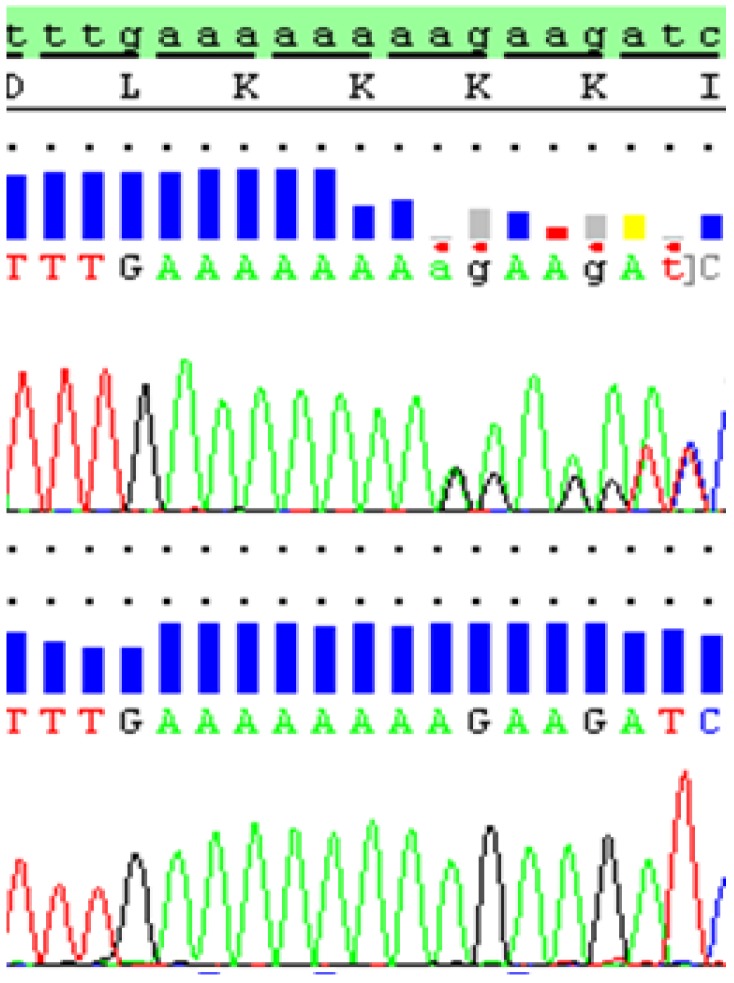
Chromatogram from Sanger DNA sequencing showing the novel c.419delA mutation in the *MCEE* gene in heterozygous form, as detected in the proband (top). The mutation causes a frameshift at Lys (K) 140 leading to a premature stop codon in the protein. Below is the normal reference sequence for comparison.

**Table 1 ijms-20-02631-t001:** Biochemical parameters reflecting status of the methionine cycle, renal function and mitochondrial metabolism with regard to methylmalonic acid (MMA). Chronic elevation of MMA is seen, together with mild MMA-uria and an associated elevation of p-propionyl-carnitine in 2016. Propionyl-carnitine was determined with a modification of the LC-MS-MS-method described by Ghoshal et al. [11], and u-methylmalonate by GC-MS-analysis of organic acids in urine [12].

Biochemical Analyses	2005 Apr, Dec	2011 Mar	2013 Mar, Oct	2015 Oct, Dec	2016 Jan, Jun
MMA (µmol/l) [<0,40]	**8.1 ^a^, 11 ^a^**	**180 ^a^**	-, **35 ^a^**	**9.9 ^b^, 16 ^b,e^**	-, **18 ^b^**
p-Hcy (µmol/l) [5,0-15]^c^, [5,0-20]^d^			**17 ^c^, 18 ^c^**	**26 ^d^, 24 ^d,e^**	-, **32 ^c^**
s-cobalamin (pmol/l) [150-650]	510, 460	420	-, 410		
s-folate (nmol/l) [7-40]	>40, -	>40			
p-propionyl-carnitine (µmol/l) [<1,3]					-, **13**
u-methylmalonate (mmol/mol creatinine) [<10]				9.0, 5.5	**10, 60**
p-creatinine (µmol/l) [<100]	-, **124**	**117**	**119, 133**	-, **189**	**159**, -

Abbreviations: MMA—methylmalonic acid; Hcy—homocysteine. ^a^ serum; ^b^ plasma; ^c,d^ different homocysteine reference intervals; ^e^ measured after treatment with high dose hydroxycobalamin.

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
