# Peer review of "MCEE Mutations in an Adult Patient with Parkinson’s Disease, Dementia, Stroke and Elevated Levels of Methylmalonic Acid"

_ijms, 2019, doi:10.3390/ijms20112631_

Round 1
Reviewer 1 Report
The title of table is not appropriate, in fact in it are reported many biochemical parameters not only about metabolic investigation (MMA, MMA-uria and p-propyonil-carnitine), I suggest to rename it.
Note : It would be interesting to know if they have considered other biochemical data indicating the metabolic acidosis of the patient as pH, bicarbonates, lactic acid PCO2 and ketones concentrations in urine sample.
The authors should briefly mention the method used to dose propionyl-carnitine, MMA in both serum and urine
Line 108-111
Authors support that hyperhomocysteinemia is associated to renal function reduced and for this reason serum values of creatinine should be added in table 1
Note : Cobalamin permits to activate two metabolic pathways, one transforms homocysteine to methionin and the other L-metylmalonyl – CoA acid to succynil - CoA. MCEE gene mutation of this patient can block the second way but not the first, thus we cannot exclude a B12 deficit.
In this study, authors report a novel mutation of MCEE gene. This mutation c.419delA (p.Lys140fs), causes a frameshift introducing a premature stop codon that has not been described before in the context of MCE deficiency. Authors have to report in the manuscript a figure of sequence analysis (chromatograms) of frameshift mutation in MCEE gene
Author Response
Response to Reviewer 1 Comments
Point 1. The title of table is not appropriate, in fact in it are reported many biochemical parameters not only about metabolic investigation (MMA, MMA-uria and p-propyonil-carnitine), I suggest to rename it.
Response 1: We have now changed the title to better reflect the content of Table 1. We have also tried to more clearly demonstrate the timeline for the different measurements.
Point 2. Note : It would be interesting to know if they have considered other biochemical data indicating the metabolic acidosis of the patient as pH, bicarbonates, lactic acid PCO2 and ketones concentrations in urine sample.
Response 2: Unfortunately, these parameters were not assessed as part of the metabolic work-up in the patient. We agree that it would have been interesting assessing these parameters, together with plasma ammonia as discussed in our manuscript, during episodes of possible metabolic stress. In particular, it would have been of interest at the time of the patient’s two strokes in 2009 and 2016.
When carefully reviewing historical medical notes, we see that in 2014 an assessment of lactate and pCO2 in blood was performed, rendering normal results. At this time the patient had problems with abdominal pain which motivated the analysis.
We have added a sentence addressing and reflecting on your comment on line 131-132.
Point 3. The authors should briefly mention the method used to dose propionyl-carnitine, MMA in both serum and urine
Response 3: We have now added a sentence in the text pertaining to Table 1, with a short description of the biochemical methods used for the determination of p-propionylcarnitine and u-MMA. This text also includes two references with regard to the used methods.
Point 4. Line 108-111: Authors support that hyperhomocysteinemia is associated to renal function reduced and for this reason serum values of creatinine should be added in table 1.
Response 4: We have now added values for p-creatinine in Table 1, to better illustrate the slowly deteriorating renal function. Furthermore, an estimated relative GFR, based on creatinine, was measured as 25 mL/min/1,73 m2 in May 2017. See point 5 for a further discussion on this topic.
Point 5. Note: Cobalamin permits to activate two metabolic pathways, one transforms homocysteine to methionin and the other L-metylmalonyl – CoA acid to succynil - CoA. MCEE gene mutation of this patient can block the second way but not the first, thus we cannot exclude a B12 deficit.
Response 5: The patient demonstrated a mild hyperhomocysteinemia in 2015 (26 µmol/l). Despite treatment with high doses of hydroxycobalamin (1 mg/day for 5 days as intramuscular injections), the homocysteine levels remained mildly elevated (24 µmol/l). Thus, we consider the likelihood of a cobalamin deficiency, contributing to the hyperhomocysteinemia seen in the patient, to be low. We have now changed the paragraph starting with line 117, hopefully better clarifying the background to why we consider the patient’s MMA elevation solely attributed to MCE deficiency, and the hyperhomocysteinemia mainly related to the patient’s renal failure.
Point 6. In this study, authors report a novel mutation of MCEE gene. This mutation c.419delA (p.Lys140fs), causes a frameshift introducing a premature stop codon that has not been described before in the context of MCE deficiency. Authors have to report in the manuscript a figure of sequence analysis (chromatograms) of frameshift mutation in MCEE gene
Response 6: A new figure has been added to the manuscript (Figure 2), showing the frameshift mutation in a DNA sequencing chromatogram.
Reviewer 2 Report
The report by Andréasson et al is a brief clinical description of a 72-year-old man with elevated methylmalonic aciduria (MMA-uria) and mutations in the MCEE gene, suggesting the MMA elevation is the result of methylmalonyl-CoA epimerase deficiency. The patient's clinical course is well described, and the likely causal role of the MCEE mutations in the MMA-uria is properly supported. The authors correctly refrain from attributing aspects of the patient's findings, beyond MMA-uria, to epimerase deficiency.
Author Response
Response to Reviewer 2 Comments
Point 1: The report by Andréasson et al is a brief clinical description of a 72-year-old man with elevated methylmalonic aciduria (MMA-uria) and mutations in the MCEE gene, suggesting the MMA elevation is the result of methylmalonyl-CoA epimerase deficiency. The patient's clinical course is well described, and the likely causal role of the MCEE mutations in the MMA-uria is properly supported. The authors correctly refrain from attributing aspects of the patient's findings, beyond MMA-uria, to epimerase deficiency.
Response 1: We thank the reviewer for the comments on our work.